# The Marine Influence Index (MII): A Tool to Assess Estuarine Intertidal Mudflat Environments for the Purpose of Foraminiferal Biomonitoring

**Frans J. Jorissen \*** , **Marie P. A. Fouet, David Singer and Hélène Howa**

UMR CNRS 6112 LPG-BIAF, Angers University, 2 Boulevard Lavoisier, CEDEX 01, 49045 Angers, France; marie.fouet@univ-angers.fr (M.P.A.F.); david.singer@univ-angers.fr (D.S.); helene.howa@univ-angers.fr (H.H.)
\* Correspondence: frans.jorissen@univ-angers.fr

**Abstract:** In this paper, we propose a marine influence index (MII), which is thought to give an integrated quantitative description of the complex of the environmental parameters controlling the foraminiferal fauna in estuarine intertidal mudflats. The MII contains three components, as follows: (1) the relative distance along the salinity gradient, (2) the emergence time relative to a reference tidal cycle, and (3) the relative importance of river outflow in the 30 days before sampling the foraminiferal fauna. Although these three parameters all have a strong relation with salinity, they also implicitly include other environmental parameters, such as the introduction of marine and continental organic matter and biota, hydrodynamic energy, or temperature. In order to show the functioning of this new index, MII is calculated for 28 stations in the Auray and Vie estuaries, for two different periods. The next step will be to compare the MII with faunal data sets. Ideally, this comparison should allow us to find strong correlations between some characteristics of the foraminiferal assemblages and the MII. If such strong correlations were indeed found, any major deviation of this relationship could then be interpreted as being due to strong anthropogenic disturbance.

**Keywords:** estuaries; biomonitoring; foraminifera; anthropogenic impact; pollution

## 1. Environmental Biomonitoring in Coastal Marine Ecosystems

Coastal areas are essential for human societies, since they host about 10% of the global population [1]. Among coastal areas, estuaries provide significant commercial and recreational benefits. They are intensively used, for fisheries, aquaculture, navigation, tourism, and many other human activities, and thereby supply huge ecosystem services. Since they constitute a physical entrance to low lying parts of the continent, they may be a vector for storm surges, often with dramatic consequences [2]. Estuaries also play a key role in the transfer of organic carbon, nutrients, and pollutants from the continent to the ocean. Thereby, they strongly influence marine life and ultimately contribute to climate change. Similar to many other transitional water bodies, estuaries are of particular interest from a biological point of view. They are among the most productive natural ecosystems in the world as they may act similar to nurseries, provide shelter to migrating fauna, and are transited by species that alternate between marine and freshwater habitats during their ontogeny.

The careful management of estuaries has been a priority for humankind, at least since the Middle Ages. Today, management not only focuses on the physical protection against floods or storm surges but is also strongly concerned with the quality of estuarine environments, as expressed by their biological, chemical, and hydrodynamic characteristics. In Europe, the Water Framework Directive obliges all of its member states to monitor the quality of their coastal and transitional water masses, to evaluate their ecological status [3] and, if necessary, to take appropriate measures to guarantee a good quality of these ecosystems. To describe the ecosystem quality, a wide range of environmental

indices have been developed, based on different characteristics, such as hydrodynamics, morphodynamics, chemical/physical properties, and especially, the biological inhabitants of the concerned ecosystems.

In this context, biotic indices based on the characteristics of the fauna/flora (such as density, diversity, and species composition) are considered to be particularly pertinent to evaluate the environmental quality. In fact, concentrations of chemical pollutants in the water column and/or in the interstitial waters of the sediment, and their comparison with reference conditions, may inform us about excess supplies resulting from anthropogenic activities. However, data on the enrichment of certain pollutants in the sediment or in the water column will neither inform us about their bioavailability, nor about their impact on the biological communities. Conversely, the density, diversity, species composition, and functional traits of the living biota may provide valuable information about ecosystem functioning, and the extent to which it may be affected by pollutants. On the one hand, the presence, absence, and/or relative abundance of certain stress-sensitive species (or functional traits) will inform us whether their tolerance levels have been surpassed or not. On the other hand, a dominance of stress-tolerant taxa, that often adopt an opportunistic life strategy, may indicate an elevated degree of environmental stress.

For marine ecosystems, the concept of Pearson and Rosenberg (1978) [4] has been pivotal. These authors showed the existence of a well-defined succession of species, with different ecological strategies, along organic matter enrichment gradients. Their conceptual model has been used to develop a large number of environmental quality indices. In the simplest form, these indices use the relative proportions of a number of faunal groups with different ecological requirements to obtain a quantitative measure of environmental quality. More elaborated indices are based on multivariate analyses of a wide range of biotic parameters [5–7].

A large number of biological groups has been proposed for these biotic indices, ranging from fishes, macro-invertebrates, macro-algae, and micro-eukaryotes. Different faunal/floral groups will give information on different spatial and temporal scales, in function of their ecology and life strategies. Consequently, the environmental quality evaluation resulting from the different indices is not necessarily identical.

Among micro-eukaryotes, foraminifera have increasingly been used over the last decennia to develop biotic indices. The international FOBIMO consortium has made significant efforts to standardize the sampling and sample treatment procedures [8]. The FOBIMO consortium has also started to adapt the AMBI-method [9], that was originally developed for macrofauna, to foraminifera. This has been achieved by attributing benthic foraminiferal taxa to different ecological groups and by testing these groups on regional datasets [10–12].

It is important to realize that in the original AMBI-method [9], and in all similar methods, including most of the foraminiferal indices of environmental quality, the species are listed in function of their response to organic matter gradients. This implicitly means that organic matter is considered to be a major, integrative, stress factor. In view of the generally coherent and robust results that are yielded by most environmental indices that are based on the Pearson and Rosenberg paradigm, this hypothesis seems to be valid in most open marine ecosystems. The underlying rationale is that in coastal marine ecosystems, excess nutrient input leads to eutrophication (increase in biological production, organic matter deposition on the seafloor, and hypoxia) [9]. In practice, nutrients and organic matter of anthropogenic origin will often be accompanied by a complex cocktail of other pollutants [13], some of which may be particularly harmful for marine life. In spite of this complexity, the indices using the response of marine biota to organic matter as an overall marker of environmental quality, often give reliable results (e.g., [14,15]). The success of this approach may be explained by the fact that most of the stress-tolerant taxa living in open marine environments are not only resistant to an increased organic matter supply (and its consequences, such as hypoxia), but also to a wide range of other stressors.

Another important aspect of most biotic indices is that they give a face value evaluation of the present-day quality of the monitored environments. These methods essentially

oversee the fact that there may be considerable variety in environmental quality in natural ecosystems. In fact, under natural conditions, some ecosystems will receive much larger nutrient and organic matter supplies than others. This is especially the case for open marine ecosystems that are influenced by river outflow, or ecosystems that are subject to deposition of large amounts of fine-grained sediments, which tend to be naturally enriched in organic matter [16,17]. In such cases, biotic indices will often indicate moderate or even bad environmental quality. This contrasts with the idea that pristine ecosystems (without any influence of anthropogenic activities), should theoretically have an optimal environmental quality. Consequently, the values of biotic indices cannot always be taken at face value but have to be compared with reference conditions. In practice, an ecological quality ratio can be determined by dividing the index values of a study site by those that are obtained for a reference site [18].

Today, the choice and description of the reference conditions remain a major subsisting problem in environmental monitoring. A fundamental question is whether the reference conditions should represent pristine conditions, comparable to the situation in pre-industrial times. Today, most of the ecosystems on Earth are, to a certain extent, affected by human activities. Since there is no possibility to rapidly overturn this situation, it is generally considered to be more realistic not to use pristine ecosystems to define reference conditions, but rather ecosystems that are judged to have a good environmental quality. In practice, usually one or more reference sites are selected, situated outside of the direct influence of the identified point sources of pollution. They are considered to represent the best environmental quality that can be found today in similar recent ecosystems and serve for comparison with the potentially polluted ecosystems under evaluation. These reference sites also represent the quality that has to be achieved by management decisions aiming to improve environmental quality.

In this context, foraminifera have a potential advantage. The preservation of their calcareous shells in the sediment record makes it possible, provided that a reliable age model can be obtained, to reconstruct the historical development of the foraminiferal community and to choose a certain time interval (for instance 1850, before the industrial revolution, or 1950) as reference conditions [19–21]. However, since important differences may exist between living foraminiferal fauna and sediment assemblages forming at the same site, the comparison of recent fauna with dead fauna that are preserved in the sediment record can almost never be straightforward. In fact, early diagenetic processes may lead to preferential loss of species with fragile tests, thereby introducing important taphonomic bias [22]. Also, important seasonal and interannual variability of living fauna may hamper the comparison with time-averaged sediment assemblages. Finally, more opportunistic species with a higher turnover ratio, often considered as pollution-tolerant, will produce more tests than stress-sensitive equilibrium species, thereby increasing their relative frequency in the sediment assemblages compared to the living fauna [23]. In spite of these limitations, foraminiferal sediment records may give valuable information about the baseline conditions in pre-industrial times [19,24].

The foregoing short introduction on biomonitoring largely relates to open marine ecosystems. In view of the encouraging results of biomonitoring indices, it is not surprising that the same methods have been applied in estuaries and in other transitional water masses. However, it became rapidly clear that, especially in estuarine ecosystems, the biotic indices very often indicated a poor or very poor environmental quality, even in cases where anthropogenic influence was apparently very limited. This negative evaluation is mainly caused by the generally low diversity and the dominance of a small number of opportunistic, stress-tolerant taxa. This observation has been termed the estuarine quality paradox [25,26]. The paradox is formed by the contradiction between apparently natural conditions, for which a good environmental quality is expected, and bad biotic index values. This contradiction results from the fact that even when they are in a natural state, most estuaries are stressed environments. This is due to the following two mains reasons:

(1)    The presence of a strong salinity gradient, from fully marine conditions at the mouth of the estuary, to brackish and freshwater in the inner part of the estuary. Few species can cope with such a huge salinity range. Most marine and freshwater species are stenohaline and can only bear minor salinity changes. It appears that estuaries are mainly inhabited by some more euryhaline marine species, whereas strictly estuarine species, living exclusively within estuaries, are rare or even non-existent [27,28];

(2)    Many environmental parameters show a high spatial and temporal variability. This is the case for salinity, which may show a huge variability on a seasonal (due to the changes in river discharge), as well as daily (weather forcing), and hourly (tidal cycles) scale. In intertidal areas, also other important ecological parameters (such as temperature, evaporation, resources, predation pressure, etc.) may vary dramatically spatially (e.g., due to different emersion times), seasonally, or even during a single tidal cycle.

The present paper envisages strategies to overcome this problem. More precisely, we will investigate how, despite the "estuarine paradox", foraminiferal assemblages can be used as indicators of environmental quality in intertidal estuarine mudflats. Our decision not to discuss all types of transitional water masses (TW's) but to concentrate on mid-latitude estuarine intertidal mudflats is motivated by (1) the fact that TW's show a very large variability in environmental factors, which makes it difficult to consider them all together, and (2) estuarine intertidal areas are particularly stressed environments because of their emergence at low tide. We will concentrate here on French Atlantic estuaries, for which excellent environmental records are available on diurnal, seasonal, and pluri-annual time scales. These estuaries are mostly tide-dominated (with a high-mesotidal to low-macrotidal regime), with the tidal currents being the main physical controlling factor. They are mostly characterized by a funnel-shaped mouth, and subtidal channels that are generally flanked by extensive intertidal mudflats.

It seems evident that in estuarine environments, an excess of anthropogenic stress can only be evaluated once the faunal response to the often severe natural stress is well known, and can be quantified. In this paper we will therefore investigate what are the main natural environmental parameters controlling the characteristics of foraminiferal communities in estuarine intertidal mudflats and how they can best be described by an easy-to-handle index. We will then test this index on two contrasting French Atlantic estuaries, for which environmental/foraminiferal datasets are available. In the companion paper of Fouet et al. (this volume) [29], we will describe these two foraminiferal datasets, and will investigate whether our description of the dominant controlling natural environmental parameters is adequate, and can explain part of the differences between the foraminiferal communities (density, diversity, and species composition) within and between these two contrasting estuaries.

## 2. Factors Controlling Faunal Distribution in Estuaries

Our ultimate aim is to develop a reliable index of environmental quality in estuaries based on the available biota. Here, we will consider this question for foraminifera, that, similar to most other organisms in estuaries show a low diversity and a strong dominance of a few opportunistic taxa. In order to develop such an index, it is of paramount importance to understand first the natural distribution of the organisms that are used in the index, and how this distribution is determined/influenced by the complex interplay of environmental parameters. In this context, by "natural distribution", we do not mean the distribution that is found under pristine conditions, but rather the distribution that is found today at sites that are, as far as we can tell, not substantially affected by anthropogenic pollution. We suppose that, even if today's ecosystems are all affected by human activities to some degree, the natural parameters are still the dominant controlling parameters of the faunal distribution. In principle, anthropogenic pollution should lead to deviations from the "natural distribution pattern". Evidently, such deviations can only be detected if the "natural distribution pattern" is perfectly known and the relations with the controlling

parameters are well understood. In view of the stress-resistant character of most of the species living in estuaries, it seems probable that only major anthropogenic perturbations can be recognized as such.

In the literature, several attempts have been made to classify estuaries (and other transitional water masses) on the basis of morphology, hydrology, characteristics of the catchment area, and climate (e.g., [28,30,31]). Although these studies present methods to classify and to compare estuaries, they do not provide the means to compare individual sites, either within the same, or in different estuaries.

An extensive discussion has taken place about the main environmental parameters controlling faunal distribution in estuaries, for macrofauna as well as for foraminifera. It is generally accepted that estuaries are among the most productive natural ecosystems in the world [32], although high turbidity due to an important sediment load may hamper/reduce primary production [27]. Despite the chronic instability of all of the other chemical/physical parameters, estuaries are not nutrient-limited. Therefore, it is generally agreed that it is not nutrients, but salinity that is by far the most important controlling factor [33,34]. The relation between salinity and estuarine fauna is often illustrated with the conceptual Remane diagram [35], which shows how the relative proportions of marine, brackish, and freshwater species change along the salinity gradient. In their excellent review, ref. [36] argued that the Remane diagram was originally proposed for the Baltic Sea, a brackish sea, and can therefore cannot be applied straightforward to estuaries. They proposed a number of refinements to the model (Figure 1) and argued that the freshwater assemblage generally has a lower species diversity than the marine assemblage and hardly penetrates into more saline waters. These authors further claimed that marine species also strongly dominate the polyhaline and mesohaline parts of the estuary, whereas exclusively estuarine/brackish species are scarce (Figure 1). In this context, on the basis of a principal component analysis of western North Atlantic estuaries, [37] demonstrated that marine species rapidly disappeared below a salinity of 24, whereas freshwater species were largely limited to salinities below four. Many studies [38,39] show that these observations are also valid for foraminifera, with the possible exception of vegetated salt marshes, which host a number of species that are typical for these ecosystems [40].

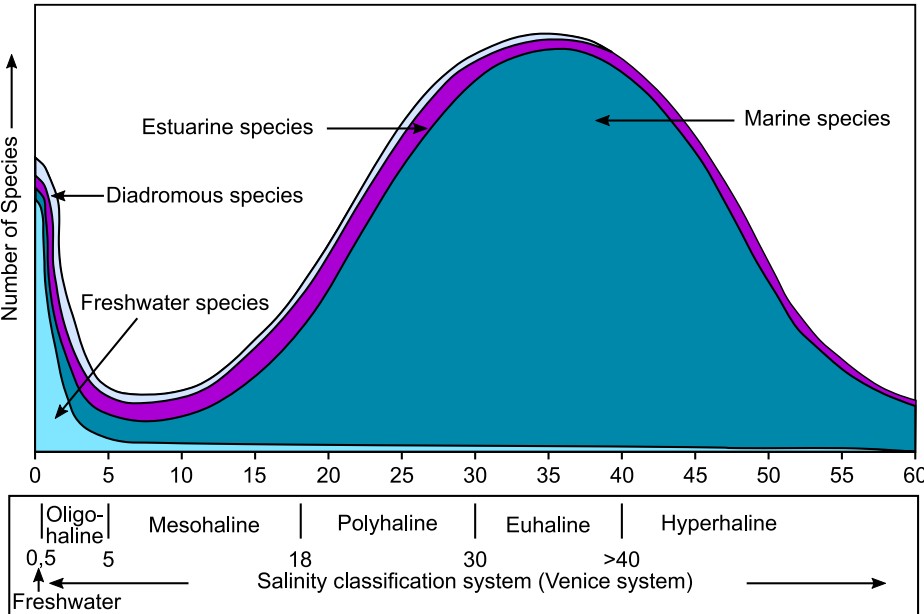

**Figure 1.** Conceptual model for estuarine biodiversity (number of species) changes covering the salinity continuum from freshwater to hyperhaline conditions (after Whitfield et al., 2012 [36]). Reprinted from Estuarine, Coastal and Shelf Science, Vol. 97, Whitfield, A.K.; Elliott, M.; Basset, A.; Blaber, S.J.M.; West, R.J., A Review of the Remane Diagram with a Suggested Revised Model for Estuaries, pages 78–90, Copyright (2012), with permission from Elsevier.

However, ref. [41] argued that it is not the absolute salinity tolerance of the various species that decides on their distribution in the estuary, but rather their tolerance of salinity variation. It seems evident that, compared to a rather stable brackish water basin, such as the Baltic Sea, the large (seasonal and diurnal) salinity fluctuations in tidally dominated estuaries must add an additional amount of stress.

The observations of similar faunal successions along transitional water transects, with sometimes very different absolute salinity values, led to the concepts of "marine vivification" [42] and "marinisation" [43]. These concepts imply that not salinity alone, but a complex of elements of marine origin (e.g., nutrients, marine flora/fauna, and hydrodynamic energy) would be the main control on the fauna living in transitional waters. Guelorget and Perthuisot (1992) [44] defined the term "confinement" as the time of renewal of these "vital elements of marine origin".

Since salt is one of the elements of marine origin, there must evidently be a correlation between salinity and "confinement". However, the absolute salinity values that are found in various parts of the estuary may be very different at different latitudes with different water budgets (different ratio between evaporation versus precipitation and river discharge). Guelorget and Perthuisot [44] concluded that it is impossible to quantify their "confinement" factor.

Debenay and co-workers applied the "confinement" concept to the foraminiferal distribution in a number of tropical estuaries [45–47]. These studies proposed a method to calculate a "confinement index", using the relative frequencies of three groups of foraminiferal species, corresponding to taxa occurring in the marine/outer estuary, middle estuary, and inner estuary, respectively. This meant that, instead of calculating a confinement index based on the physical properties of the estuary and then comparing the calculated index values with the data on the faunal distribution, Debenay and co-workers did exactly the opposite. In our opinion, this approach is not entirely free from circular reasoning; the high percentages of species that are considered typical of a limited marine influence (while living in the inner parts of the estuary), will unavoidably lead to high values of the confinement index there, indicative of limited marine influence.

Unfortunately, it is often far from easy to obtain reliable salinity measurements in an estuary. Especially in macrotidal estuaries as both the subtidal and intertidal areas experience huge temporal salinity variation on an hourly scale, as salinity rises with the incoming tide and decreases with the outgoing tide. A reliable picture of this pattern can only be obtained by continuous data logging, which is only possible for a limited number of points. Even when such a continuous record is available [48], it is not evident what precise salinity value(s) should be used to describe the ecosystem. For the intertidal areas, which may emerge every day for many hours, when they are affected by evaporation and/or precipitation, the problem is even larger. Therefore, ref. [41] suggested characterizing the salinity regime by measuring the range between the mean low-tide salinity and the mean high-tide salinity. This may be a solution for the subtidal part of the estuary, but not for the intertidal areas, where it is basically impossible to measure salinity at low tide.

Summarizing, there is a consensus that salinity and/or marine factors related to salinity, are the main controls of the faunal characteristics in estuaries. Together, these parameters can be regrouped under the term "marine influence". This term not only refers to the introduction of salt, but also of marine organic matter, flora, and fauna, and especially, tidal energy. Due to the tidal energy, this "marine influence" also partly controls factors, such as sediment grainsize, water column turbidity, oxygen concentration, and residence times of continental waters, in the estuary. The latter factor may have important consequences regarding the use of river-born nutrients, primary production within the estuary, the bottom, and the water column oxygen contents, as well as the geochemical nutrient and organic matter recycling in the estuarine superficial sediments [49].

It appears, therefore, that a complex of "marine influence" factors, which largely exceeds salinity alone, constitutes the main control of estuarine biota. Many, if not all of these factors (salinity, strong tidal currents, and hypoxia) may be causes of natural stress.

In intertidal areas, additional stress results from emergence and faunal exposure during low tide, leading to major variations of temperature, salinity, and predation pressure. In general, a decrease in marine influence leads to an increase in stress for marine species.

In this paper, we will propose a new marine influence index (MII), with which we attempt to describe the cumulative effect of the above-mentioned stress factors affecting estuarine foraminifera. If the MII correctly describes the main factors controlling the foraminiferal assemblages, its values should show a strong correlation with the main faunal characteristics. If so, this index could serve to predict the characteristics of the fauna under natural conditions. Clear deviations from the natural distribution pattern, as predicted from such an MII, would then be due to factors that are not included in the MII, such as anthropogenic pollution. Here, we will develop this marine influence index and test it on two estuaries on the French Atlantic coast. In the companion paper of Fouet et al. [29], this MII will be tested on the foraminiferal data of the same estuaries.

## 3. Ecology of Estuarine Foraminifera

Estuarine foraminiferal assemblages have been abundantly studied. Excellent reviews are given in [39,50–52]. It is important to note that many of the earlier studies in estuaries were based on total assemblages (i.e., counts of all foraminiferal tests, of living, as well as dead, individuals) and not only on living fauna, as we propose here. Until recently, the use of total assemblages was privileged because they were considered to give an average picture, not biased by the often important seasonal and spatial variability of the living assemblages [50]. Although the total assemblages may be useful for comparison with fossil samples (for instance for the detailed reconstruction of ancient sea levels), their use for biomonitoring purposes has several pitfalls. In fact, the time slice recorded by a total assemblage may be very different between sites, due to the very different sedimentation rates found within single estuaries. In some cases, the time presented by a total assemblage may largely exceed the period that is characterized by anthropogenic impact. In other cases, bottom erosion, occurring frequently in estuaries, may erode recent sediments and lead to a mixture of living and much older dead foraminifera. Debenay et al. (2000) [50] argued that the potential bias (between living and total fauna) that is introduced by post-mortem transport and taphonomic losses may have been overestimated. We disagree with this opinion and think that the higher diversity that is often recorded in estuarine total assemblages (compared to living fauna) is the direct consequence of a landward transportation of tests of marine species by tidal currents. The presence of such allochthonous taxa will mainly inform us about the hydrodynamics, and not about the other local environmental conditions.

Unlike total fauna, living fauna reflect the environmental conditions prevailing during the weeks to months before sampling. Therefore, we judge that for biomonitoring studies, the use of living assemblages should be mandatory [8], in spite of the complexity created by important seasonal, interannual, and spatial variability of the assemblages.

Similarly, for macrofauna, the salinity gradient within the estuary, from freshwater in the inner part to marine salinity at the mouth, appears to be the dominant controlling parameter of the foraminiferal spatial distribution. This contrasts with open marine environments, where food availability and oxygen concentration, and in shallower areas also temperature, and the nature of the substrate, are the dominant controlling parameters [51,53]. Most foraminiferal taxa are stenohaline, limiting their habitat to the outer estuary, where diversity is maximal. Much fewer species seem to be adapted to the mesohaline and polyhaline conditions found in the middle estuary. Finally, foraminifera become scarce in the oligohaline upper estuaries, where thecamoebians flourish [39]. In this paper, we will not consider salt marshes, which host a highly specific fauna, usually strongly dominated by agglutinant taxa [54].

Many other parameters, other than salinity variation alone, will influence the ecology of estuarine foraminifera. The temperature, food availability, oxygen concentration, sediment characteristics, and hydrodynamics will all control to some extent the foraminiferal

density, diversity, and species composition. These factors may be especially important when their values are close to the upper or lower tolerance limits of individual species.

However, it appears that the natural distribution of foraminifera is foremost determined by salinity, or, more precisely, by salinity variation. In intertidal mudflats, which form the subject of this paper, biological stress caused by emergence at low tide, leading to abrupt changes in salinity and/or temperature, appears to be an important secondary factor. Because of the multiscale temporal and spatial variability of salinity in estuaries, this parameter is not synchronized with foraminiferal life cycles, and neither snapshot nor long-term salinity measurements are suitable to describe the influence of this factor on foraminiferal fauna. The optimal description of this major controlling environmental factor at each study site remains to be a major challenge that we will consider in this paper.

Salinity in estuarine waters depends on the mixing of marine and continental waters. Several factors play a role here, as follows:

- **Tidal influence** will diminish with increasing distance from the sea and will vary along the lunar-solar tidal cycles. The tides largely control the estuarine circulation and the associated biological processes. The hydrodynamics of the rising and falling tide generates turbulence and causes vertical mixing between fresh and marine waters. Depending on the local tidal regime, river discharge, and the morphology of the estuary, estuaries may be weakly to strongly stratified. The seasonal changes in river discharge and the dynamics of the salt-wedge intrusion also affect biological production [27]. For example, in a seasonally stratified estuary, the presence of the salt wedge in spring led to increased nutrient recycling and phytoplankton blooms [49]. The tidal range will, together with the elevation of the study site, also determine the emergence time of the various parts of the tidal flats;
- **River discharge**, that dilutes the sea water intruding by tidal processes into the estuary, may show major seasonal and interannual variability, depending on the climate regime. Consequently, the sites that are located in the inner estuary may show major seasonal salinity changes, from almost-marine to almost-freshwater;
- **Local precipitation** is a second source of fresh water, which will especially affect the salinity of the emerged mudflats during low tide;
- **Groundwater inflow** may be an important additional source of fresh water, which is often difficult to estimate;
- **The morphology of the estuary** may facilitate or hamper the propagation of the tidal wave and/or of the marine waters in the estuary.

Together, these factors not only decide on salinity, but also determine the extent of "marine influence" in the different parts of the estuary. The term "marine influence" not only includes salinity, but also introduced marine organic matter and marine biota (as adults, larvae, or propagules). This is important, since marine organic matter tends to be more labile than continental organic matter [55]. Marine biota may serve as food or may fuel the colonization of the estuary by marine species, during (short) periods with suitable environmental conditions. Essentially, increased marine influence is the expression of a higher ratio between the volume of sea water entering the estuary and the freshwater supplies (through river runoff, groundwater input and precipitation). Increased marine influence may result in longer freshwater residence times in the estuary. This may lead to a more substantial utilization of river-born nutrients within the estuary [49]. At low tide, the extent of "marine influence" will rapidly decrease, especially in the case of intertidal estuarine mudflats that are subjected to strong precipitation and/or evaporation.

## 4. Quantifying the Main Parameters Controlling Foraminiferal Distribution in Estuarine Intertidal Mudflats

Here, we will try to quantify the main physical and hydrological parameters summarized under the term "marine influence", since they appear to be responsible for most of the natural environmental stress in intertidal estuarine mudflats, to which the foraminiferal community is exposed.

Next, we will investigate the possibility to assemble these factors into an easily applicable marine influence index (MII).

### 4.1. Position of the Sampling Point along the Salinity Gradient

Salinity appears to be the foremost factor controlling the ecology of estuaries, particularly in the case of tide-dominated estuaries [56], as suggested by the Remane diagram (Figure 1). The factor "Salinity" includes absolute salinity values, as well as the range of salinity variation. Because of strong temporal variability, it is already a challenge to measure these parameters in subtidal estuarine environments. This becomes almost impossible in intertidal mudflats, which are emerged at low tide. We suggest that the most integrative parameter related to salinity is the extent of the salt intrusion into the estuary. This is the point until which the salt-water wedge enters the estuary at incoming tide. Beyond this point, the water is permanently fresh, even during the highest spring tides. However, the physical tidal influence may be felt further inland, in the fluvial estuary, where the incoming tidal wave may partly block the outflowing fluvial waters. Therefore, the upper limit of the estuary is often placed more inland than the salt intrusion [57]. In our study, that focuses on the influence of the inflowing marine waters on foraminiferal fauna, we will not consider the areas beyond the salt intrusion. The information concerning the salt intrusion is available for almost all French estuaries [58], but it is not clear when exactly these observations have been made. This is important, because the salt wedge intrudes much farther into the estuary in the dry season compared to the wet season. If data on the salt intrusion are not available, field measurements at high spring tide (during the dry season) can define its upper limit.

The relative position of the sampling point between the mouth of the estuary and the point of farthest salt intrusion can be used to characterize the position of the sampling point on the salinity gradient. This relative distance can be defined as follows:

$$X/S \tag{1}$$

where X is the distance (measured along the channel axis, in km) from the mouth of the estuary (defined on the basis of geomorphological criteria), and S is the length of the salt intrusion (in km). This ratio describes the relative position of the sampling point along the salinity gradient, irrespective of absolute salinity values, tidal range (micro- to macrotidal), and the strength of the salinity gradient (which varies in function of the seasonal dilution by fresh water). As such, this measure should be more robust than absolute salinity values, not only because these are challenging to obtain, but also since absolute salinity values may show large latitudinal variation, in function of different climate regimes.

It can be envisaged that the extent of the salt-water intrusion does not diminish linearly with increasing distance from the mouth. In fact, it can be affected by the presence of mechanical obstacles that may cause strongly localized reductions in salt-water inflow. These obstacles may be of the following three different kinds:

(1) **Turtuosity**: Sea water will have more difficulty entering a meandering estuarine channel than a rectilinear estuary, especially in the case of a strongly curved shape of the channel. For this reason, we propose to measure the relative position of the sampling point with respect to the real distance from the inlet, considering all of the eventual curves of the main channel (i.e., definition of X in Equation (1);

(2) **Abrupt narrowing of the estuary or channel sills:** Narrowing, or sills of geological origin, will cause major constraints for salt water supplies to the upper parts of the

estuary. Such "bottlenecks" will accentuate the non-linearity of the horizontal salinity gradient, and thereby shorten the length of the salt intrusion (i.e., S in Equation (1)). This factor, which is difficult to quantify, should be at least partly integrated in the decrease in marine influence with relative increasing distance from the mouth of the estuary;

(3)　**Manmade physical obstacles**: On the French Atlantic coast, many of the small and medium sized estuaries are closed upstream by a dam provided with a sluice. These dams/sluices are used to prevent salt water from intruding into the fluvial estuary beyond these dams. They are generally closed during rising tides in order to protect fields that are under cultivation from soil salinization. They open during some of the falling tides in order to regulate the outflow of riverine freshwater into the estuary. This is done to keep fluvial waters available for irrigation purposes or to flush the lower parts of the estuary, to remove clay deposits in the navigation channel. When the dam/sluice is closed, the salinity tends to be elevated until the dam is reached, with freshwater being present immediately landward of the dam. As a result, these dams artificially limit the salt intrusion, truncating abruptly the salinity gradient from marine to freshwater. In such cases, we think that the (theoretical) natural salt intrusion should be used to define the relative position of the sampling point on the salinity gradient, and not the observed salt intrusion (until the mechanical obstacle). This natural salt intrusion (before the instalment of the manmade barrier) can be assessed by a careful study of the thalweg topography; normally salt water enters the estuarine valley until the point where the relief starts to increase, often abruptly.

*4.2. Altitude/Emergence Time*

The altitude of the sampling site has a major impact on the ecology of the foraminiferal fauna living in intertidal areas, for the following three main reasons:

(1)　The elevation of a site may change the position of the foraminiferal assemblage with respect to the vertical salinity gradient, due to haline stratification. In highly stratified estuaries, the salt-water wedge enters the estuary below the outflowing superficial freshwater layer. If present, this vertical gradient will change on diurnal and seasonal timescales in response to incoming and outgoing tide, and to fluctuations in river discharge. Precipitation may be a secondary reason for the development of a vertical salinity gradient, even in the case of estuaries with a homogenized water column. Whatever the exact reasons and salinity values, at higher elevation, the foraminiferal fauna will generally be confronted with lower salinity;

(2)　In combination with the tidal range, the elevation largely influences the time of emergence at low tide. The twice-daily emergence of the tidal mudflats exposes the intertidal habitats to major changes of moisture, temperature, and salinity (due to evaporation and/or precipitation). In fact, in the intertidal realm, the time of emergence seems to be a much more important stress factor than slight salinity changes due to the different positions of the sampling sites on the vertical salinity gradient. For macrofauna, emergence also leads to changes in predation pressure, as the contribution of fish decreases while predation by shore-birds and mud-snails becomes more important [59]. Little information is available for predation on foraminifera [60], although there are indications for selective predation by gastropods [61]. Together, these changes represent a substantial recurrent stress;

(3)　Hydrodynamic energy will progressively diminish from the estuarine channel to the most elevated parts of the intertidal mudflats. The tidal currents may displace foraminiferal fauna, leading to an underrepresentation of small species and introduction of allochthonous taxa in the intertidal realm. Hydrodynamics will also control sediment grain size and organic matter availability (as organic particles are preferentially bound to clays).

The question is how to describe these different effects of habitat elevation with a single quantitative proxy. The absolute altitude, i.e., the elevation above the lowest astronomical tide (chart datum), that is by definition a fixed value, cannot take into account the fluctuating ecological stress that is imposed by the emergence time variability. The emergence time depends mainly on the local tidal range. This is not only determined by the global tidal cycles (14 days and annual), but also by the spatial variability of the tidal range within the estuary. Due to the propagation of the tide wave in a complex estuarine geometrical system, a tidal asymmetry is produced (mainly by bottom friction and wave refraction), leading to amplitude and phase differences, from the river mouth to the inner estuary [62].

For these reasons, we suggest not to use absolute altitude, but rather the emergence time, that can be considered as an integrative factor for the environmental controls listed above.

The emergence time (ET, expressed as a fraction of total time) can be calculated for each sampling point on the basis of absolute altitude and the tidal tables, which are provided by SHOM for all French harbors [63]. We decided to use the mean spring tide data (in France: tidal coefficient of 90) to calculate a reference emergence time for all of the sites. The choice of conditions typical of a specific part of the 14-day tidal cycle is necessary to be able to compare the sites at different altitude. Next, a spring tide situation was chosen, because during spring tide, every part of the intertidal mudflats is at least immerged for part of the day. Conversely, during neap tide, the lower parts of the mudflats will be immerged all day, whereas the uppermost parts of the mudflats may never be reached by the rising tide.

It is important to realize that the relative emergence time (ET) is not determined by the global tidal cycles but by the local tidal range (micro- to macrotidal regime, up- to downstream location), so that the tidal range parameter does not have to be further considered. Moreover, the relative emergence time also partly expresses the hydrodynamics of a site. Short relative emergence times mostly concern the intertidal areas that are located close to the channel axis, where tidal dynamics are maximal. Inversely, the sites with long relative emergence times will generally be far from the channel axis and will experience a much lower hydrodynamic energy.

A detailed description of the methods that were used to estimate the absolute altitude and emergence time has been added as Supplementary Materials.

### 4.3. Importance of Fluvial Discharge

The extent of fluvial discharge is of paramount importance for the intrusion of the marine wedge. A large discharge volume will not only diminish the distance of the saltwater intrusion, but will also lead to a smaller volume of salt water entering the estuary. In fact, some very large rivers, such as the Congo, have such large outflow volumes that the intrusion of the salt wedge becomes almost impossible year round, even at high tide [57]. Fluvial discharge will, also in our mid-latitude region, show large seasonal variability, with important maxima in the winter/early spring and a minimum in the summer. Generally, salinity (marine influence) largely decreases in the inner parts of the estuaries during winter runoff maxima. This may lead to a temporary strong diminution or even the complete disappearance of foraminiferal fauna [64]. For all major French rivers, and for most major rivers worldwide, mean annual discharge volumes, as well as detailed runoff data, are available (http://www.hydro.eaufrance.fr/, accessed on 23 September 2021). However, such detailed data may be more difficult to obtain in other regions. It is important to evaluate which precise data (of what exact period) are relevant for the survey of benthic faunal constituents.

Foraminifera are supposed to have a rapid response to environmental change, as is shown by many examples of strong seasonal faunal variability, in response to changes of runoff volume and precipitation [54]. This rapid response is the direct result of their short life cycles (estimated from six months to two years [51]). Therefore, it is not relevant to compare the living foraminiferal community with year-averaged river discharge data. It seems more judicious to compare the faunal data with the conditions in the period imme-

diately before sampling. Consequently, we decided to characterize the fluvial discharge regime by averaging the values of the 30 days before sampling.

Next, the volume of river discharge has to be related to the size of the estuary, in order to evaluate its impact on the introduction of marine waters into the estuary. In their comparison of tidal estuaries along the Atlantic coast of Europe, Middelburg and Herman [65] used (year-averaged) freshwater residence times in order to characterize the nine studied estuaries. They found that in general, river-dominated estuaries have short average freshwater residence times (up to two weeks), whereas strongly tidal-dominated estuaries have much longer average river water residence times (one to three months). Residence times are important, because in the case of longer freshwater residence times, estuarine environments are more easily fueled with marine nutrients, and riverine nutrients are recycled within the estuary, leading to increased biological productivity [64].

There is extensive literature on the estimation of freshwater residence times in estuaries, that essentially shows that the calculation of such residence times is complex [66–68]. We suggest an easier way to compare the volume of river discharge with the size of the estuary (the total amount of water), notably to calculate for every sampling station the ratio between the average freshwater discharge (in the 30 days before sampling of the foraminiferal assemblages) and the cross-section of the estuarine channel at the sampling point at the mean high spring tide, as follows:

$$\text{RRO (relative river outflow)} = \text{RD (river discharge)}/\text{CS (estuarine cross-section)} \qquad (2)$$

Since most estuaries become wider in a seaward direction, this ratio will have the tendency to decrease from the inner to the outer parts of the estuary, although bottlenecks may lead to non-linear changes in the RRO.

*4.4. Distance from the Main Estuarine Channel*

There is a large difference between the estuaries with limited width and narrow tidal flats, and the estuaries with large tidal mudflats, sometimes with entire embayments emerging at low tide. In both cases, the main part of the inflowing and outflowing water volume passes through an often narrow main channel, where the hydrodynamic energy is maximal. The hydrodynamic energy will decrease with increasing distance from the main channel. Consequently, tidal and fluviatile currents are stronger in the upper parts of narrow mudflats that are close to the main channel, than in the upper parts of very wide mudflats, which will only be attained at high tide. The hydrodynamic energy is probably also a major factor for the transport of tests and propagules of foraminifera and may, therefore, be an important vector for the (re)colonization of estuarine habitats.

It appears that the decrease in hydrodynamic energy with increasing distance from the main channel is at least partly represented by the altitude/emergence time factor ET. In fact, the width of the estuary is also taken into account in the relative river outflow factor RRO, since it is the main parameter that is used for determining the surface area at the sampling sites. The RRO factor should therefore also be indicative, at least of the hydrodynamic energy, due to outflowing river water, which will be higher in estuaries with a high RD/CS ratio, in which a large volume of river water flows out through a comparatively narrow estuary. We think, therefore, that it is not necessary to add a specific term for this factor to our index of marine influence.

## 5. Defining the Marine Influence Index (MII) for Estuarine Intertidal Mudflats

The first three components that are discussed above together determine the level of marine influence in intertidal estuarine mudflats, that is considered as the main controlling factor of the foraminiferal fauna living there. They can be assembled into a marine influence index (MII) in the following way:

$$\text{MII} = \alpha\left(1 - \frac{X}{S}\right) \times \beta\left(1 - \text{ET}\right) \times \gamma\left(1 - \sqrt{0.04 \times \frac{\text{RD}}{\text{CS}}}\right) \qquad (3)$$

where:

X = real distance from the sampling site to the mouth of the estuary, following the main estuarine channel. X is measured between the mouth of the estuary and a line starting from the sampling site, perpendicular to the channel axis.

S = distance of the natural salt intrusion into the estuary; the observed value when no obstacle is present, a theoretical value based on the thalweg topography when a manmade obstacle is present.

ET = emergence time, expressed as a fraction of a tidal cycle, calculated for mean spring tide conditions (in France: tidal coefficient 90).

RD = river discharge in the 30 days before sampling (in $m^3 s^{-1}$).

CS = surface area of the estuarine cross-section at high tide (in $m^2$).

Details about the calculation of S, ET and RD/CS are given in the Supplementary Materials.

The equation is written in such a way that in case marine influence is total, all three of the individual parameters, as well as MII, are one. The terms α, β and γ are constants, translating the respective weight of each of the three factors. The values of these constants have to be determined in future, on the basis of a comparison of faunal and environmental parameters in a wide range of estuarine mudflats.

In this equation, both (1-X/S) and (1-ET) are defined as normalized, dimensionless ratios, varying from 1.0, equivalent to the maximum marine influence, to 0.0, no marine influence. This is not the case for the RD/CS ratio. We decided to normalize this factor by comparing its values to the one obtained in the Loire River estuary at Cordemais, at 28 km from the mouth of the estuary. This site coincides with the administrative limit between the marine and fluvial domain, determined by the limit between mesohaline (5–18 PSU) and oligohaline (0.5–5 PSU) realms. The farthest point to which the salty water can intrude is located at 60 km from the mouth of the estuary, slightly upstream of the town of Nantes.

The Loire River was chosen as a reference riverine system because it is the longest river in France (1013 km long), draining a catchment basin of 118,000 $km^2$, and evolved in the same climate regime as the studied estuaries on the Atlantic margin. Despite the fact that the tidal range at the entrance is 5.4 m (at spring tide) and the tide head (the farthest point upstream where a river is affected by tidal fluctuations) is located 97 km from the mouth, the estuary of the Loire River is the one with the strongest river influence along the French Atlantic Coast. The Loire estuary is affected by a mean annual freshwater discharge of 854 $m^3$/s, varying from 10-year floods of 5200 $m^3$/s to summer low water flows of 140 $m^3$/s. Consequently, the freshwater residence time varies from 3 to 30 days [69]. At the Cordemais site, the fluvial estuary is about 750 m wide, with a cross-section of about 3400 $m^2$. This means that, when considering the mean annual discharge, RD/CS is 25 cm/s.

For all of the other estuaries on the French Atlantic margin that have a lower freshwater input than the Loire estuary, a normalized RRO (relative river outflow) value, varying between 0.0 and 1.0, is calculated. First, the RD/CS value is divided by 25 (the reference value calculated for the Loire at the Cordemais site). In this way, all of the sites in all estuaries can be compared on the basis of a normalized RRO value. Finally, in order to give more weight to differences at the lower part of the 0.0 to 1.0 scale, we use the square root of the normalized RRO value for the MII calculation. In the remaining part of this paper, we will apply this marine influence index on the Auray and Vie estuaries, for which we also dispose of the foraminiferal records. We want to investigate how the MII differs (1) between estuaries with different characteristics, (2) between the different parts of the estuaries, and (3) between different periods, with ample, or almost no river runoff.

## 6. Applying the Marine Influence Index (MII) to the Auray and Vie Intertidal Estuarine Mudflats

In order to show how this marine influence index describes spatial and temporal differences between and within different estuaries, we will apply the concept on two contrasting French estuaries on the Atlantic coast. In the companion paper of Fouet et al. [29], it is investigated whether the MII can be used to explain the faunal variability that was observed in these estuaries.

### 6.1. Auray Estuary

The Auray estuary (Figure 2a) is a ria, a drowned river valley with a surface area of about 12 km². It is part of the much larger Morbihan Gulf, a marine bay, and is a tide-dominated estuary. The entrance of the estuary is 950 m wide. Two small rivers flow out into the estuary, the Loc'h, with a mean annual discharge volume of 2.72 m³/s, and the Bono, with a mean annual discharge volume of 1.52 m³/s. Together, these two rivers drain a 60 km long catchment area of 324 km². Total freshwater residence time is estimated at about two months, underlining the fact that the estuary experiences a strong marine influence. The estuary has a meso- to macrotidal regime, with a tidal range of 4 m at the mouth during mean spring tides. The salt wedge intrudes until a distance of 20 km from the mouth of the estuary. During the 2019 survey, in the context of the European Framework Directive, the chemical properties of Auray estuary were evaluated as good, whereas the ecological state was evaluated as average [70]. The latter judgement was based on an indicator using fishes. Another indicator, based on macro-algae, indicated a good environmental quality. These results suggest that, in spite of an apparently strong anthropogenic pressure (oyster culture and recreational navigation), the living biota, including the foraminiferal community, are not subjected to severe stress of anthropogenic origin.

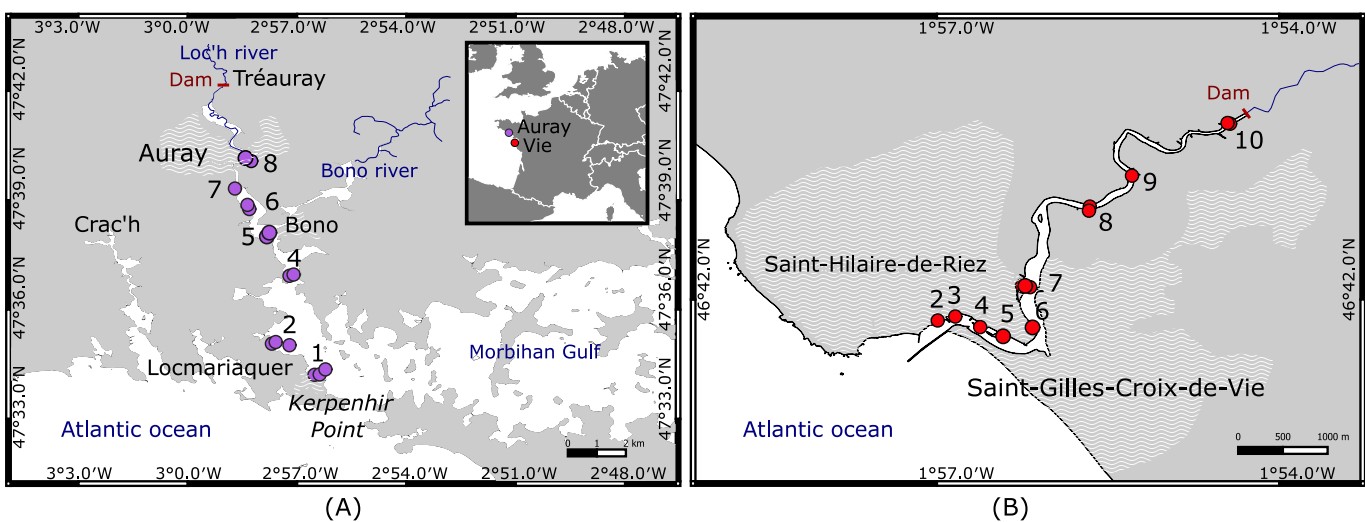

**Figure 2.** Schematic maps of the Auray (**A**) and Vie (**B**) estuaries on the French Atlantic coast with location of the sampling sites. Urbanized areas are indicated with a light grey shading.

In the Auray estuary, at seven sites, for a total of 15 stations, foraminiferal fauna were sampled from 10 to 12 July 2019 (Figure 2). The main characteristics of these sites are listed in Table 1. When several sampling stations were selected at different altitudes at a same site, the station at the highest elevation systematically has the suffix A, whereas stations B and C are positioned at progressively lower altitude.

**Table 1.** Environmental characteristics of the 15 sampling stations at 7 sites on intertidal mudflats in the Auray estuary. Altitudes refer to the French chart datum. The general methodology of the calculation of MII is explained in Section 5; a more detailed explanation of the individual components is given as Supplementary Materials.

| Station | Distance to Sea (X, in km) | Normalized Distance to Sea (1-X/S) | Distance Perpendicular to Channel Axis (m) | Altitude (m) | Emergence Time (%) | Normalized Emergence Time (1-ET) | River Discharge/ Cross-Section (RD/CS, m/s) | RRO (Relative River Outflow) | MII ($\alpha$, $\beta$ and $\gamma$ Set at 1.0) |
|---|---|---|---|---|---|---|---|---|---|
| 1A | 2.8 | 0.86 | 1150 | 2.52 | 42.3 | 0.58 | 0.03 | 0.97 | 0.48 |
| 1B | 2.8 | 0.86 | 1055 | 1.29 | 19.7 | 0.80 | 0.03 | 0.97 | 0.67 |
| 1C | 2.8 | 0.86 | 880 | 1.38 | 21.1 | 0.79 | 0.03 | 0.97 | 0.65 |
| 2A | 4.9 | 0.75 | 1450 | 4.50 | 88.6 | 0.11 | 0.03 | 0.97 | 0.08 |
| 2B | 4.9 | 0.75 | 1380 | 4.20 | 77.2 | 0.23 | 0.03 | 0.97 | 0.17 |
| 2C | 4.9 | 0.75 | 880 | 1.38 | 21.1 | 0.79 | 0.03 | 0.97 | 0.57 |
| 4A | 8 | 0.60 | 275 | 3.30 | 55.0 | 0.45 | 0.15 | 0.92 | 0.25 |
| 4B | 8 | 0.60 | 210 | 1.37 | 21.1 | 0.79 | 0.15 | 0.92 | 0.43 |
| 5A | 10.4 | 0.47 | 300 | 2.20 | 36.9 | 0.63 | 0.09 | 0.94 | 0.28 |
| 5B | 10.4 | 0.47 | 60 | 1.21 | 17.8 | 0.82 | 0.09 | 0.94 | 0.37 |
| 6A | 11.8 | 0.40 | 200 | 2.50 | 42.3 | 0.58 | 0.18 | 0.92 | 0.21 |
| 6B | 11.8 | 0.40 | 50 | 1.49 | 24.3 | 0.76 | 0.18 | 0.92 | 0.28 |
| 7 | 13.2 | 0.33 | 100 | 1.31 | 20.4 | 0.80 | 0.30 | 0.89 | 0.24 |
| 8A | 15 | 0.24 | 100 | 2.30 | 38.5 | 0.61 | 0.54 | 0.85 | 0.13 |
| 8B | 15 | 0.24 | 65 | 1.29 | 19.7 | 0.80 | 0.54 | 0.85 | 0.17 |

The seven sampling stations are positioned between 2.8 (station one) and 15.0 km (station eight) from the inlet of the Morbihan Gulf, at the Pointe de Kerpenhir. Since the salt wedge is observed until the Moulin de Tréauray, 19.8 km from the mouth of the estuary, the normalized distance from the mouth (1-X/S; column two in Table 1) varies from 0.86 (station one, outer estuary) to 0.24 (station eight, inner estuary). Samples were taken between 1.29 and 4.5 m above French chart datum and consequently, the percentage of emergence (ET) varies from 17.8% (station 5B) to 88.6% (the highest station 2A), corresponding to the normalized emergence times (1-ET) of 0.82 and 0.11, respectively.

The RRO values were calculated (see Supplementary Materials for methodology) based on the river discharge volume recorded between 10 June and 9 July 2019, the 30 days before taking the foraminiferal samples (which are studied by Fouet et al. [29]). During this period, the discharge of the Loc'h River gradually decreased from 3.92 m$^3$/s on 12 June 2019 to 0.55 m$^3$/s on 9 July 2019, with an average value of 1.50 m$^3$/s. This value is about 55% of the yearly average of 2.72 m$^3$/s. For the Bono River (average yearly discharge 1.52 m$^3$/s), no monthly data are available, but the yearly trend should be very similar, so that we decided to estimate the discharge during the 30 days before sampling at 0.84 m$^3$/s, using the same ratio as observed in the Loc'h River between the yearly average and the study period.

The obtained normalized RRO (relative river outflow) values vary from 0.85 (the most inland, site eight) to 0.97 (site one, closest to the mouth), showing that, compared to the Loire estuary, which serves as a standard (see Section 5), riverine influence is very weak in the Auray estuary.

In Table 1, the constants $\alpha$, $\beta$, and $\gamma$ of the MII index (Equation (3)) are all set to one, in order to give an equal weight to the three parameters (X/S, ET, and RD/CS). The values of these constants have to be determined at a later stage by comparison with faunal patterns (considered as "natural") in a wide range of estuaries. In Figure 3, the MII values, which vary from 0.67 (station 1B) to 0.13 (station 8A), have been plotted in a 3D diagram. The obtained isolines of MII connect points, which should experience a similar degree of marine influence, should therefore host comparable fauna.

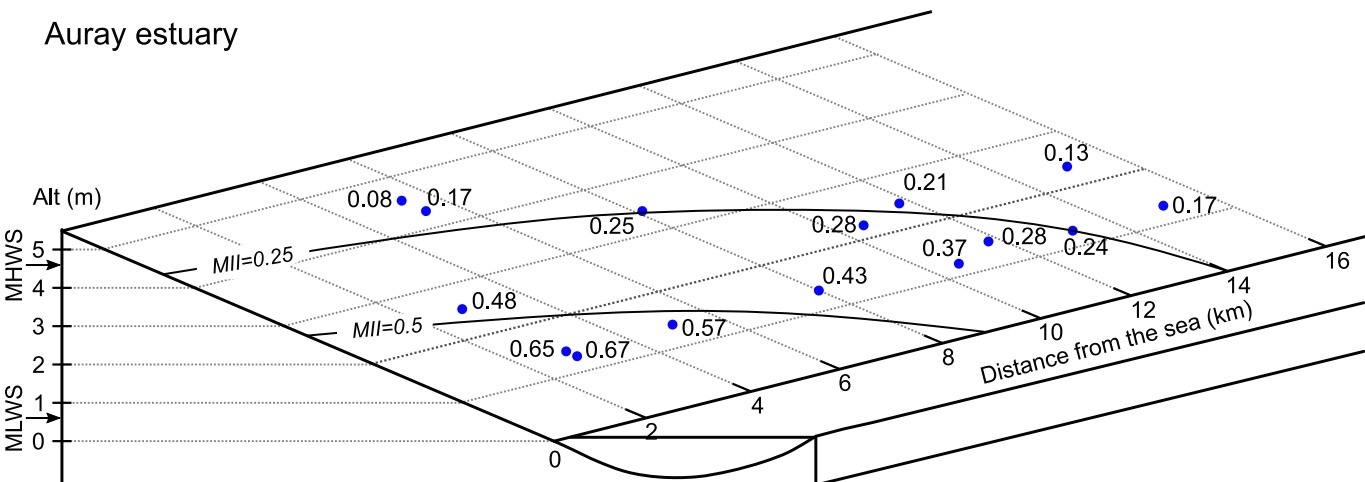

**Figure 3.** Overview of sampling sites in the Auray river estuary with values of the MII index, based on runoff data between 10 June and 9 July 2019. Altitudes refer to the French chart datum. See text for further explanation. Horizontal and vertical axes correspond to the distance to the sea and the elevation of the sampling stations, respectively. Tidal reference levels are indicated on the *Y* axis; MHWS: Mean high water springs; MLWS: Mean low water neaps.

*6.2. Vie Estuary*

The Vie estuary (Figure 2B) is a typical lowland estuary. The Vie estuary receives freshwater from the Vie River, that is 63.5 km long, has a surface area of 0.95 km$^2$, and drains a catchment area of 84 km$^2$. The entrance of the estuary is about 200 m wide. Today, the salt wedge penetrates the estuary until a dam with a sluice at 8 km from the entrance. However, a detailed observation of topographical maps suggests that before the construction of this dam, salt water penetrated the estuary until about 16.8 km inland. In fact, the estuary shows a meandering pattern through the low coastal plain, with a nearly horizontal thalweg from the entrance of the estuary to the village of St. Maixent-sur-Vie, where it starts to rise abruptly. In our calculations of the normalized distance to the sea, we have used this putative value of 16.8 km.

The Vie River has a mean annual discharge volume of 1.18 m$^3$/s. The total freshwater residence time is estimated at about one week, showing that the Vie estuary is strongly river-dominated. The estuary has a meso- to macrotidal regime, with a tidal range of about 4 m at the mouth. During the 2019 survey, in the context of the European Framework Directive, the biological status was evaluated as good, whereas the chemical status was not investigated [70].

Since our chemical analyses only show minor enrichment of Cu in some small harbors, and no enrichment of other toxic metals (Fouet et al. [29]), we conclude that the Vie estuary is not affected by severe anthropogenic stress. This contrasts with the strong urbanization and intensive recreational pressure in the lower part of the estuary (Figure 2B).

In the Vie estuary, nine sites with a total of 13 sampling stations were selected for our study of the foraminiferal community (Figure 2B, Table 2), and were sampled from 23 and 24 October 2018. The main characteristics of the sampling stations are listed in Table 2. The 13 sampling stations are positioned between 0.4 (station two) and 7.7 km (station 10) from the mouth of the estuary. The normalized distance from the mouth (1-X/S) varies from 0.97 (station two) to 0.53 (station eight). Samples were taken between 1.15 and 4.86 m above French chart datum and consequently, the percentage of emergence (ET) varies from 7.6% (station 8B) to 77.9% (the highest station 10A), corresponding to the normalized emergence times (1-ET) of 0.92 and 0.22, respectively.

**Table 2.** Environmental characteristics of the 13 sampling stations at 9 sites on intertidal mudflats in the Vie estuary. Altitudes refer to the French chart datum. The general methodology of the calculation of MII is explained in Section 5; a more detailed explanation of the individual components is given as Supplementary Materials.

| Station | Distance to Sea (X, in km) | Normalized Distance to Sea (1-X/S) | Distance Perpendicular to Channel Axis (m) | Altitude (m) | Emergence Time (%) | Normalized Emergence Time (1-ET) | River Discharge/Cross-Section (RD/CS, m/s) | RRO (Relative River Outflow) | MII (α, β and γ Set at 1.0) |
|---|---|---|---|---|---|---|---|---|---|
| 2 | 0.4 | 0.98 | 185 | 3.35 | 48.32 | 0.52 | 0.01 | 0.98 | 0.49 |
| 3 | 0.5 | 0.97 | 65 | 2.00 | 27.21 | 0.73 | 0.01 | 0.98 | 0.69 |
| 4 | 0.7 | 0.96 | 90 | 1.82 | 24.49 | 0.76 | 0.01 | 0.98 | 0.71 |
| 5 | 0.9 | 0.95 | 170 | 2.67 | 37.84 | 0.62 | 0.01 | 0.98 | 0.58 |
| 6 | 1.9 | 0.89 | 50 | 1.44 | 16.44 | 0.84 | 0.02 | 0.97 | 0.72 |
| 7A | 2.6 | 0.85 | 185 | 4.82 | 76.47 | 0.24 | 0.02 | 0.97 | 0.19 |
| 7B | 2.6 | 0.85 | 140 | 4.18 | 62.25 | 0.38 | 0.02 | 0.97 | 0.31 |
| 7C | 2.6 | 0.85 | 85 | 1.61 | 20.41 | 0.80 | 0.02 | 0.97 | 0.65 |
| 8A | 4.6 | 0.73 | 35 | 4.21 | 62.91 | 0.37 | 0.05 | 0.96 | 0.26 |
| 8B | 4.6 | 0.73 | 15 | 1.15 | 7.64 | 0.92 | 0.05 | 0.96 | 0.64 |
| 9 | 5.4 | 0.68 | 10 | 1.22 | 11.03 | 0.89 | 0.06 | 0.95 | 0.58 |
| 10A | 7.7 | 0.54 | 40 | 4.86 | 77.92 | 0.22 | 0.06 | 0.95 | 0.11 |
| 10B | 7.7 | 0.54 | 10 | 2.24 | 31.08 | 0.69 | 0.06 | 0.95 | 0.36 |

The RD/CS values were calculated on the basis of river discharge volume during the 30 days before the taking of the foraminiferal samples, from 23 September to 22 October 2018. During this period, the discharge varied between 0.001 and 0.002 m$^3$/s for 23 September to 7 October 2018, and a maximum of 0.089 m$^3$/s was measured on 21 October 2018. The average discharge during these 30 days was only 0.021 m$^3$/s, which is exceptionally low for this period (Figure 4). The obtained normalized RRO values vary from 0.98 (site one) to 0.95 (the inner estuary site 10).

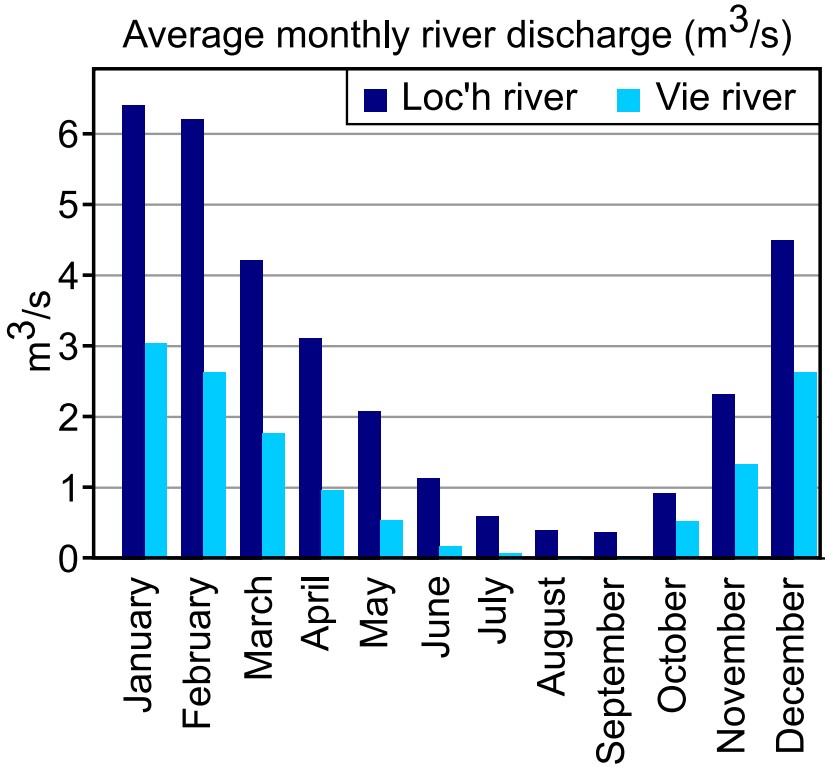

**Figure 4.** Average monthly discharge volumes for the Loc'h and Vie rivers (average of 49 years of observation). Source: http://www.hydro.eaufrance.fr/, accessed on 23 September 2021.

In order to calculate the MII values, the scaling constants α, β, and γ were all set to 1.0, so that the three parameters constituting the MII have been given equal weight. In Figure 5, the MII values, which vary from 0.72 (station six) to 0.11 (station 10A), have been plotted in a 3D diagram. The obtained isolines of the MII connect points experiencing a similar degree of marine influence, which should potentially host comparable fauna.

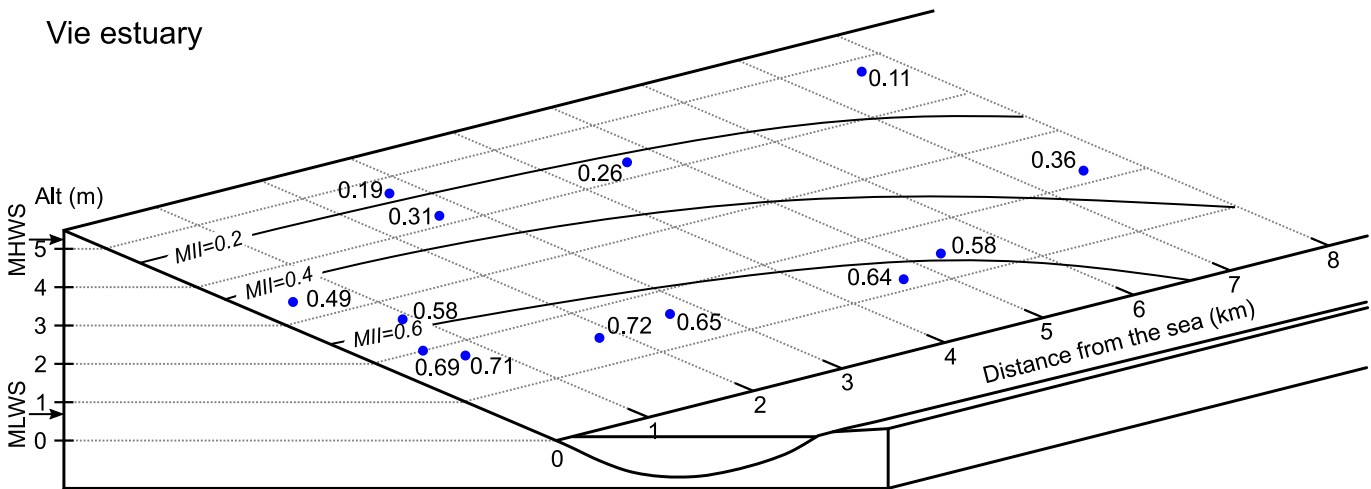

**Figure 5.** Overview of sampling sites in the Vie estuary with values of the MII index, based on runoff data between 10 June and 9 July 2019. See text for further explanation. Horizontal and vertical axes correspond to the distance to the sea and the elevation of the sampling stations, respectively. Tidal reference levels are indicated on the *Y* axis; MHWS: Mean high water springs; MLWS: Mean low water neaps.

*6.3. Comparison of the Two Estuaries, Impact of Seasonal Variation of River Discharge*

When we compare the MII values for the Auray and Vie estuaries, it can be noticed first that the range of index values is very comparable, from 0.67 to 0.13 for the Auray estuary, compared to 0.72 to 0.11 for the Vie estuary. Nevertheless, the values for the position of the sampling point on the salinity gradient are higher in the Auray estuary. This is due to the fact that we did not sample landward of the dam at 8 km from the mouth, that artificially blocks the salt-water influence upstream. In both of the estuaries, riverine influence was minimal in the 30 days before sampling, as is expressed by the very low RD/CS values that vary from 0.03 to 0.54 in the Auray estuary and from 0.01 to 0.06 in the Vie estuary. These values are very small compared to the (year-averaged) value of 25.0 calculated for the Loire estuary at Cordemais. They show that in both of the estuaries, the river discharge was very low in the 30 days before sampling, with values of 2.34 and 0.02 m³/s for the Auray and Vie estuaries, respectively.

In order to better show the influence of river discharge on the MII, we calculated for all of the sites of both of the estuaries the MII values on the basis of the year-averaged runoff volumes, of 4.29 and 1.19 m³/s for the Auray and Vie estuaries, respectively (Tables 3 and 4).

**Table 3.** Auray estuary; MII calculated for observed river charge in June–July 2019 and for average river discharge.

| Auray Estuary | River Discharge/ Cross-Section (RD/CS, m/s) | RRO (Relative River Outflow) | MII (α, β and γ set at 1.0) | River Discharge/ Cross-Section (RD/CS, m/s) | RRO (Relative River Outflow) | MII (α, β and γ set at 1.0) |
|---|---|---|---|---|---|---|
| Station | | June–July 2019 | | | Average River Runoff | |
| 1A | 0.03 | 0.97 | 0.48 | 0.06 | 0.95 | 0.47 |
| 1B | 0.03 | 0.97 | 0.67 | 0.06 | 0.95 | 0.66 |
| 1C | 0.03 | 0.97 | 0.65 | 0.06 | 0.95 | 0.64 |

Table 3. *Cont.*

| Auray Estuary | River Discharge/ Cross-Section (RD/CS, m/s) | RRO (Relative River Outflow) | MII (α, β and γ set at 1.0) | River Discharge/ Cross-Section (RD/CS, m/s) | RRO (Relative River Outflow) | MII (α, β and γ set at 1.0) |
|---|---|---|---|---|---|---|
| Station | June–July 2019 | | | Average River Runoff | | |
| 2A | 0.03 | 0.97 | 0.08 | 0.05 | 0.96 | 0.08 |
| 2B | 0.03 | 0.97 | 0.17 | 0.05 | 0.96 | 0.16 |
| 2C | 0.03 | 0.97 | 0.57 | 0.05 | 0.96 | 0.57 |
| 4A | 0.15 | 0.92 | 0.25 | 0.29 | 0.89 | 0.24 |
| 4B | 0.15 | 0.92 | 0.43 | 0.29 | 0.89 | 0.42 |
| 5A | 0.09 | 0.94 | 0.28 | 0.16 | 0.92 | 0.28 |
| 5B | 0.09 | 0.94 | 0.37 | 0.16 | 0.92 | 0.36 |
| 6A | 0.18 | 0.92 | 0.21 | 0.34 | 0.88 | 0.21 |
| 6B | 0.18 | 0.92 | 0.28 | 0.34 | 0.88 | 0.27 |
| 7 | 0.30 | 0.89 | 0.24 | 0.54 | 0.85 | 0.23 |
| 8A | 0.54 | 0.85 | 0.13 | 0.98 | 0.80 | 0.12 |
| 8B | 0.54 | 0.85 | 0.17 | 0.98 | 0.80 | 0.16 |

Table 4. Vie estuary; MII calculated for observed river charge in June–July 2019 and for average river discharge.

| Vie Estuary | River Discharge/ Cross-Ection (RD/CS, m/s) | RRO (Relative River Outflow) | MII (α, β and γ Set at 1.0) | River Discharge/ Cross-Section (RD/CS, m/s) | RRO (Relative River Outflow) | MII (α, β and γ Set at 1.0) |
|---|---|---|---|---|---|---|
| Station | October 2018 | | | Average River Runoff | | |
| 2 | 0.01 | 0.98 | 0.49 | 0.29 | 0.89 | 0.45 |
| 3 | 0.01 | 0.98 | 0.69 | 0.50 | 0.86 | 0.61 |
| 4 | 0.01 | 0.98 | 0.71 | 0.34 | 0.88 | 0.64 |
| 5 | 0.01 | 0.98 | 0.58 | 0.32 | 0.89 | 0.52 |
| 6 | 0.02 | 0.97 | 0.72 | 1.40 | 0.76 | 0.57 |
| 7A | 0.02 | 0.97 | 0.19 | 0.88 | 0.81 | 0.16 |
| 7B | 0.02 | 0.97 | 0.31 | 0.88 | 0.81 | 0.26 |
| 7C | 0.02 | 0.97 | 0.65 | 0.88 | 0.81 | 0.55 |
| 8A | 0.05 | 0.96 | 0.26 | 2.98 | 0.65 | 0.18 |
| 8B | 0.05 | 0.96 | 0.64 | 2.98 | 0.65 | 0.44 |
| 9 | 0.06 | 0.95 | 0.58 | 3.40 | 0.63 | 0.38 |
| 10A | 0.06 | 0.95 | 0.11 | 3.40 | 0.63 | 0.08 |
| 10B | 0.06 | 0.95 | 0.36 | 3.40 | 0.63 | 0.24 |

For the Auray estuary, the MII values change only marginally, but in the Vie estuary, the MII becomes much lower, with values below 0.5 in the whole inner part of the estuary. These values, which were calculated for year-averaged runoff volumes, show the large difference between the two estuaries very well. The Auray estuary is a ria, widely open into the marine Morbihan Gulf, and is, therefore, more tide-dominated than the Vie estuary, which has a narrow meandering course and is a strongly river-influenced estuary. Of course, in the Vie estuary, river-influence is much less prominent in the dry periods with minimal runoff volumes, which existed during the 30 days before our sampling in October 2018. If we would have used the maximal average river discharge values (3.12 m$^3$/s, average value for January, based on a 49-year record), the MII values would be below 0.5 in the whole estuary, with the exception of sites three and four, close to the mouth. This suggests that in winter, conditions in a large part of the Vie estuary are unfavorable for foraminifera.

## 7. Discussion and Conclusions

In this paper, we have investigated the dominant factors determining the natural distribution of foraminifera in intertidal estuarine mudflats. Next, we have attempted to assemble these factors in a single index of marine influence (MII), which ideally should give an overall characterization of the natural conditions of intertidal estuarine sites. The aim of this MII is to make it possible to compare different sites in the same estuary, and/or to compare sites in different estuaries.

Three environmental parameters have been selected to calculate the MII, because we hypothesized that they provide the best description of the environmental context in which the foraminiferal community develops, as follows:

(1)   The relative position of the sampling point on the gradient between fully marine water and freshwater;

(2)   The relative emergence time of the sampling point at the mean spring tide conditions, expressed a percentage of the total length of a tidal cycle. This parameter is closely related to the absolute altitude (with respect to the level of the lowest astronomical tide) of the sampling point;

(3)   The relative importance of freshwater outflow, which is determined by dividing the average discharge volume in the 30 days before sampling by the cross-section area of the estuary at each sampling point.

All three of the parameters are clearly related to salinity, which is generally thought to be the dominant factor controlling estuarine fauna, but implicitly also include other controlling factors, such as the introduction of marine and continental organic matter and biota, temperature, and hydrodynamic energy. In view of their short life cycles (estimated between six months and two years, ref. [51]), and their high reactivity to environmental changes, a time lag of about two to four weeks between the environmental conditions and the foraminiferal response seems appropriate. For this reason, we calculate the average river discharge for the 30 days preceding the sampling of the foraminiferal fauna.

For the moment, we have not given different weights to each of the three theoretical factors constituting our MII. In order to be able to attribute such a relative weight, we added the constants $\alpha$, $\beta$, and $\gamma$ into Equation (3), but for the time being, these were set at one, giving equal weight to the three variables. The absolute values of $\alpha$, $\beta$, and $\gamma$ will be determined in future, on the basis of a large-scale comparison of MII and the characteristics of the foraminiferal communities, in a large number of estuaries in different seasons. A multivariate analysis should make it possible to quantify the respective contribution of the three main components of the MII to the multivariate model, and thereby, to determine the values of the weighing constants.

In order to investigate whether this marine influence index (MII) correctly describes the distribution of foraminiferal faunas in intertidal estuarine mudflats, a comparison with faunal data is necessary. A first attempt was made in the companion paper of Fouet et al. (this volume) [29]. However, in this paper only a part of the large range of environmental variability was considered, especially with respect to river outflow, which was minimal at the time of sampling. In order to appreciate the efficiency of the MII as an overall descriptor of foraminiferal habitats in intertidal estuarine mudflats, a more exhaustive comparison with field studies is necessary. Such a more important study should ultimately also allow us to determine the values of the three constants.

The aim of the MII is to serve as a tool to predict the composition of foraminiferal assemblages in intertidal estuarine mudflats, in particular the relative frequencies (at different sites) of marine, estuarine, and freshwater species, and of stress-tolerant taxa. It should be kept in mind that the MII presented here has been developed to quantify the complex environmental parameter "marine influence", that appears to be main factor that is responsible for the natural environmental stress characterizing intertidal estuarine mudflats, to which the foraminiferal community is exposed. It is evident that factors other than "marine influence" may affect foraminiferal communities as well. These may be physical/chemical (e.g., temperature, oxygen concentration, and substrate grain size) or biological factors

(predation and competition). However, most of these additional factors are in some way related to the distance to the sea and the emergence time as well, so they are already partly included in the MII. The MII is therefore expected to explain a large part of, but not all of the characteristics of the foraminiferal community in natural conditions.

A comparison between the observed faunal characteristics and those expected on the basis of the MII should make it possible to detect the sites at which the fauna are strongly impacted by anthropogenic pollution. At such sites, the faunal composition should show major differences with the composition expected on the base of the MII, which cannot be explained by other natural factors.

Summarizing, the marine influence index (MII) that is proposed here, is meant to be a first step in the development of a foraminiferal index of environmental quality, specific for estuarine intertidal mudflats. A phase of calibration of the index, by comparison with faunal data sets is now necessary in order to investigate whether the MII can indeed provide a reliable description of the complex of environmental parameters controlling the characteristics of the foraminiferal community in such environments.

**Supplementary Materials:** The following supporting information can be downloaded at: https://www.mdpi.com/article/10.3390/w14040676/s1, Table S1: Tidal reference levels for St. Gilles Croix de Vie (Vie estuary) and four sites in the Auray estuary: Port Navalo (mouth of the estuary), Locmariaquer (2.8 km from the mouth), Fort-Espagnol (8.0 km from the mouth) and Auray (15.0 km from the mouth). Table S2: RD/CS and RRO (relative river outflow) calculated for four sites in the Loire estuary, using an average yearly discharge volume (RD) of 854 m$^3$/s. Table S3: RD/CS and RRO (relative river outflow) calculated for four sites in the Loire estuary, using the discharge volume (RD) of 193 m$^3$/s, the average value measured during low water conditions in October 2019. Table S4: Comparison of MII, Altitude, Distance to the sea, Normalised distance to the sea and Salinity, for the Auray and Vie estuaries. Figure S1: Schematic representation of the duration of emergence at a sampling point (arbitrary chosen at 2.6 m altitude) during a mean spring tide cycle, for the site Auray in the upper Auray estuary. In this example, the duration of emergence is 6 h 14'30'', whereas the tidal cycle takes 12 h 19'. The corresponding reference emergence time will be 50%. Figure S2: Map of the Loire estuary with the position of the four localities used here. Figure S3: Relation between measured salinity and distance to the sea for the Auray estuary (in blue, July 2019) and Vie estuary (in orange, June 2020). Figure S4: Relation between measured salinity and normalised distance to the sea for the Auray estuary (in blue, July 2019) and Vie estuary (in orange, June 2020). Figure S5: Bottom water salinity (in the main estuarine channel) in Bono (black line) and Locmariaquer (grey line) between March 2006 and March 2008. After Diz et al. [48].

**Author Contributions:** Conceptualization, F.J.J.; writing, F.J.J., M.P.A.F., D.S. and H.H.; visualization, D.S.; supervision and project administration, F.J.J. All authors have read and agreed to the published version of the manuscript.

**Funding:** This research was funded by the OFB (Office français de la biodiversité) and the University of Angers, grant number 3976-CT_RD_AMI_18_SURV_FORESTAT.

**Acknowledgments:** We are very grateful for the numerous discussions with colleagues that allowed us to develop the ideas presented in this paper.

**Conflicts of Interest:** The authors declare no conflict of interest. The funders played no role in the design of the study, in the writing of the manuscript, or in the decision to publish the results.

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
