# Peer review of "The Marine Influence Index (MII): A Tool to Assess Estuarine Intertidal Mudflat Environments for the Purpose of Foraminiferal Biomonitoring"

_water, doi:10.3390/w14040676_

Round 1

Reviewer 1 Report

estuarine mudflats

This manuscript details the computation of an indicator of marine influence called MII based on 3 simple metrics :

-Relative distance from the estuary mouth

-Relative Emergence time

-Importance of fluvial discharge on cross section

This manuscript first explains in details the choice of each three metrics and the general context justifying the need of such an index.

Trying to find a proxy of marine influence to explain the spatial distribution of benthic communities in estuaries is not new (I would suggest to also refer to the works of Galvan et al. 2010, Hume et al 2007, Blanchet et al 2014) Indeed, I guess that every community ecologist working with estuarine fauna and flora communitues attempted to define such a proxy. However this paper proposes to adress this issue in a simple manner that could be easily used in many different temperate estuaries. The author provide interesting justifications regarding the choice they made to build their index of marine influence (MII). The paper is interesting and well written but there are, in my opinion, several drawbacks that should be adressed to improve the relevance of the paper :

First, this paper only describes the rationale and the computation of the index but does not actually test the index. This is very frustrating since at the end of the paper one needs to know if it actually works and how much of estuarine fauna variations is explained by this index, otherwise it is worthless as every unpublished attempts that one has performed with ones data. Instead, the authors talk about a companion paper that is going to test this index with foraminifera data (Fouet et al., same issue). In the Jorissen et al parper, the index is applicated to two case studies : the Auray estuary and the Vie estuary. For both estuaries, the values of the MII is computed at several locations within this estuary. The variation of the values of the index are then measured in situations of high and low discharges. The index is shown to reflect differences between the two types of estuaries (one is a ria and one is river-dominated estuary). But still, the reader is not provided with any clues on the ability of this index to actually explain faunal patterns : the MII is not tested, it just varies. I think that one way to cope with this issue would be

(i) to compute this index for more than the two estuaries under scrutiny here. Indeed, one does not need to actually sample other estuaries to model the behaviour of the MII with emergence duration. This would in particular allow to challenge the choice of the author of using the Loire at Cordemais as a reference for the RD/CS ratio. This would also give an idea of the applicability of this index to larger estuaries such as the Loire estuary.

(ii) To challenge the index with actual salinity data, even if the index measures more than salinity level and salinity variations. It would show that this index correctly reflects the salinity variations with season

Another issue is that the discussion focuses on abiotic envrionemental factors and how they affect foraminifera assemblages. However, the authors should put more attention on biotic factors that also explain community pattern. This could include at least inter-species relationships within the foraminifera community, inter-species relations with other benthic organisms which modify the local environment and could explain foraminifera pattern and the lack of fit with the MII (see further comments).

I have another issue regarding the way X, « the real distance from the sampling site to the mouth of the estuary following the thalweg » is determined : unfortunaltely the use of « geomorphological criteria » (as indicated on line 421) is not sufficient because (1) defining this limit is user-dependant and (2) the salinity levels at the « geomorphological » entrance varies greatly among estuaries. I think that the author should instead define a point where (e.g.) the salinity reaches 33 during high discharge. This issue must be (at the very least) discussed. This is why the MII should be challenged with salinity data.

The same issue regards the definition of the upstream limit of salt intrusion to define the S-value. The authors state (line 457 to 459 and lines 718 to 722) that the intrusion of the salt wedge can be estimated using topographical maps : « the topography is nearly horizontal from the entrance of the estuary to the village of St Maixent-sur-Vie, where it starts to rise abruptly ». I honestly don’t understand how the author performed this and I am not sure of the topography they are speaking of : is it the river bed topography or the land topography ? In both case authors should be more precise and provide convincing references to support the way they defined this limit. Again using salinity data would be useful here.

Finally, the index uses three parameters (alpha, beta and gamma) that could give different weights to the three components of the MII. They are set to one by default… The autors should however give some idea on how one should try to define these values in the future.

There are some minor issues regarding the manuscript :

Lines 239 to 243 is a paragraph that is similar to the one on lines 228 to232- please fix this issue

Line 301-302 « Clear deviations from the natural distribution pattern as predicted from such a MII could then be ascribed to anthropogenic stress » - I think that one should write instead that it also «  could be ascribed to other ecological factors not included in the index which includes anthropogenic stress ».

Line 333 « This constrasts with open marine environments , where food availability is usually the dominant controlling parameter »- please provide several references that demonstrates this pattern and/or correct this sentence if it only concerns the deep areas.

Line 575-computation of the MII: (1) please make sure that emergence time ET is indeed a % (with a 0-100 scale) and not a proportion (with 0 to 1-scale) ; (2) the gamma symbol in equation (3) looks like a y, please check that it is an actual gamma symbol.

As a conclusion, I think that there are too many unknown regarding the accuracy of this index (values of alpha, beta, gamma ; important issues regarding how one actually locates the upstream and downstream limits of an estuary) and there are no evidence to support that this index could actually work. Despite the fact that the paper, the idea and the justifications of the choices performed by the authors make sense and are well justified I think that the paper does not provide sufficient evidence of teh accruracy of this index unless the authors provide at least evidence that their index at the very least reflects salinity values and fluctuations (see point (ii)) and/or prove that the MII can be easilly used in other estuaries and reflect seasonnal variations of river discharge.

Cited references

Galván, C., Juanes, J.A., Puente, A., 2010. Ecological classification of European transitional waters in the North-East Atlantic eco-region. Estuarine, Coastal and Shelf Science 87, 442-450.

Hume, T.M., Snelder, T., Weatherhead, M., Liefting, R., 2007. A controlling factor approach to estuary classification. Ocean & Coastal Management 50, 905-929.

Blanchet H., Gouillieux B., Alizier S., Amouroux J.M., Bachelet G., Barille A.L., Dauvin J.C., de Montaudouin X., Derolez V., Desroy N., Grall J., Gremare A., Hacquebart P., Jourde J., Labrune C., Lavesque N., Meirland A., Nebout T., Olivier F., Pelaprat C., Ruellet T., Sauriau P.G., Thorin S. (2014) Multiscale patterns in the diversity and organization of benthic intertidal fauna among French Atlantic estuaries, Journal of Sea Research, 90, 95-110, doi: 10.1016/j.seares.2014.02.014.

Reviewer 2 Report

Foraminifera are protists that have been increasingly used in bio-monitoring. Indices of ecological quality status are based either on taxonomic diversity or on the sensitivity of species to ecological stress. Attempts to develop an index for the intertidal zone have not been very successful because the intertidal zone is a naturally stressed biotope inhabited by a limited number of stress-tolerant taxa. The authors come up with a completely novel approach. They suggest to develop first an index that quantifies the environmental gradient in estuarine intertidal settings (in this paper) and then to test whether this index explains the foraminiferal distribution (in another paper). In case of success, this will be the reference concordance of the environment and the foraminiferal fauna. A disruption of the concordance will be interpreted as anthropogenic impact and then quantified into an index of ecological quality status. Neither the authors nor I know whether the idea will work after all. However, this is a new idea that gives new insights, and I do think that the paper should be published.

To explain what is the rationale behind their index, the authors overview concepts and notions of the science that deals with marine coasts, including ecology, oceanography, surface-water hydrology, and geomorphology. It is a tremendous task to synthesize all this knowledge. The paper is logical and readable, which is a great success already. So many layers of knowledge are merged that the paper cannot be easy reading. I feel my main task here is to suggest how the text can be clarified in places and made easier to understand.

I recommend a moderate revision.

General comments

  1. Emergence time:

The target auditorium, of course, vary in its scientific background. I comprehend it is a challenge to choose what to explain and what to assume to be broadly known. Your statement “the relationship between altitude and emergence time is not linear” is followed by a very short explanation, which will not be easily understood. Elaboration is needed. You may say that the tidal dynamics are quasi-sinusoidal, or you may, preferably, choose to show harmonics for the studied estuaries. You calculate emergence times using local tide tables. Tide tables are calculated for a certain reference point. The tidal range will decrease landwards in a constricted estuary like yours. The time phase will lag landwards. The asymmetry of the tidal harmonic will increase landwards. When you present later your own data on emergence time, specify the reference points you have. I do not understand whether you correct your emergence time values for the above variation. If not, say explicitly the correction is skipped at this early stage of the project. 

  1. Intertidal zonation:

Many readers are used to dividing the intertidal zone by Mean Spring Tide and Mean Neap Tide, whereas you show only exposure time. If possible, please add the MST and MNT levels to you illustrations. Or explain explicitly that you do not apply these levels because they are uncertain in constricted estuaries like those you investigate.

  1. Predation:

(1) Predation on intertidal macrofauna is mentioned several times in different chapters using essentially the same words. I doubt numerous repetition is needed. (2) The reader is expected to know what is meant, but I think an explanation should be given. (3) Line 475 reads as if predation pressure decreases at higher intertidal elevations. The authors possibly imply predation by wading birds. The trend I suppose will be opposite if predation pressure is imposed by sea dwellers such as flounders or starfish. (4) I expect to read next about predation on intertidal foraminifera, but this never comes. If there are no reliable data, say there are no reliable data.

  1. Fluvial Influence:

The name Fluvial Influence is misleading (formula 2). The term implies that there is a signal that weakens continuously from the river to the sea. Firstly, it comes into a logical conflict with section 4.1. and its formula 1, which have already taken the distance between the river and the sea into account. Secondly, the formula of Fluvial Influence does not incorporate any variable related to the river-to-sea distance. The only variable attached to the sampling station is the area (square meters) of the cross-section of the estuary at the station. Therefore, Fluvial Influence will decrease seawards only when the estuary is geomorphologically funnel-shaped (and this is mentioned somewhere in the text). If there is a bottleneck, for example, a bedrock sill or sand bars at the mouth, then Fluvial Influence values will be grater at the mouth than in the inner estuary. Perhaps, a better term would be River/seawater volume ratio or something like that. The formula will still be applicable to funnel-shaped estuaries only, but it will be easier for the reader to understand what is meant. See also my specific comments.

  1. Structure and clarity:

The authors review a large body of literature, and the structure of the paper is not standard. They give an overview of bio-monitoring, an overview of estuarine environment, and an overview of estuarine foraminiferal ecology. Then the authors explain how they compose the new index. Finally, they apply the index to environmental data collected in two estuaries. There are conclusions at the end. This unusual structure can be kept, I think. Most of text is readable and logical. Only chapter 6 is problematic. The methods should be clearly combined into a separate section or at least a separate paragraph. The language needs to be checked for clarity and conciseness. Copy-and-paste has been used extensively to exchange text between sections 6.1. and 6.2. This redundancy would better be reduced. See also my specific comments.

  1. The aim:

Aims keep popping up. I suggest you formulate the main aim in the first chapter. If a certain chapter afterwards has own aim, then make it obvious that this is a subordinate aim that belongs to this chapter only.

  1. Figures:

If the volume allows, I would recommend to show 2D salinity profiles along the two studied estuaries, for low tide and high tide. Consider also adding a graph showing tidal harmonics for the estuaries. Show the tidal reference points on the maps. There are suggestions in the specific comments to Figs. 3 and 5. 

  1. Literature cited:

The number of entries in the citation list is within 60. Considering the diversity of lines of research covered in the paper, there has been effort to sort out references and to keep the list to a minimum. They are principal publications mostly. I have not cross-checked citations in  the text and the reference list. This paper deals with quantifying the environment only. The second half of this research project deals with foraminiferal assemblages and has been submitted as a separate paper (Fouet et al.). No single value from that manuscript is used here, and yet I see a score of references to that unpublished work. This is way too many. Mentioning a couple time will do. When cited authors are the subject of a sentence, you say either “Debenay et al. (2000) [44] argued…”  or just “[44] argued…” Please check the journal guidelines for the correct usage. I have seen three versions in the text: Debenay and co-workers, Debenay and co-authors, and Debenay et al. Please unify.

Specific comments

* [Title] There are no foraminiferal assemblages in this paper, and the title should be changed. Maybe the following version is better. The Marine Influence Index (MII): a tool to quantify the environment of estuarine intertidal mudflats for the purpose of foraminiferal bio-monitoring

* [14] The Abstract should contain a line on your MII calculations.  

* [14-17] A half of the Abstract is on what is planned to be done in another paper. Very unusual. I do not like it, but it is acceptable. If the authors prefer to stick to this version, I would trust the authors’ choice.

* [18-19] The last sentence of the Abstract, which contains a reference to an unpublished paper, would better be omitted or at least modified. For example, it can be said that foraminifera were sampled at the same stations, and this faunal part of the project is being published separately. Omit references in the Abstract.

* [92-96] The text is going and going without references and sounds like a discussion inserted into the introduction. I understand this chapter is supposed to give an overview what has been achieved and what remains poorly understood in biomonitoring of coastal ecosystems. Please avoid placing your arguments and opinions in this chapter.

* [152-156] references are needed.

* [170-173] you were going to explain why you focus on French Atlantic estuaries and ignore others, but instead you are describing the physical setting in French Atlantic estuaries. Perhaps, you may say there is an excellent environmental record for the estuaries on diurnal, synoptic, and multi-year time scales. 

* [179-184] These lines are difficult to follow. “We will test this index on two contrasting French Atlantic estuaries for which environmental/foraminiferal datasets are available.” I am not sure I understand the meaning.  What is the aim of this paper? Does the following lines describe the aim of another paper, not this one? Would it be possible to be more specific on which “part of the differences” is expected to be explained?

* [186] An aim again. Is this the aim of this paper, a collaborative aim of the two companion papers, or the aim of a big project in the future?

* [218 ff] “of western north Atlantic.” Perhaps North should be capitalized.

* [202-232] You present an extended overview of ecological stress caused by decreased salinity. There have been studies in the Baltic (I cannot recollect a good reference though) showing that when seawater is diluted to 8 psu or lower, the ratio of the ions is not marine anymore, and this is an additional source of ecological stress. Decide yourself whether you need this in your overview. 

* [228, 239] the paragraph is repeated.

* [251] Omit NaCl. You speak of the standard marine cocktail of ions, not of Na and Cl only.

* [257-267] The chapter is on factors controlling faunal distribution, while you speak of distribution itself in this paragraph. I doubt this paragraph belongs here.

* [297-305] This is mostly a repeat of what was said in the last paragraph of section 2 [179-184]. The wording is much better here. You may choose to use this paragraph instead of that one.

* [333-334] Not quite clear whether you speak here of salinity as an environmental factor or, as you stated earlier, of a combination factors, which act together along the salinity gradient.

* [362] weakly to strongly perhaps

* [363-365] The sentence needs to be more specific.

* [388] river-born

* [387] “This may lead to a more substantial utilisation of river born nutrients within the estuary and a higher production of microphytobenthos in intertidal areas.” You may well be correct, but this is not obvious and needs references. Or omit.

* [392 and elsewhere] Something is wrong with the heading. If you already know what “the main parameters controlling foraminiferal distribution are,” then why do you undertake this study? Perhaps you want to say Important abiotic parameters of aestuarian habitats… or something like that.

* [394] This is the third or forth instance of an aim being formulated. If this aim belongs to chapter 4 than say it explicitly.

* section 4.1. This section, I think, is very well written. Everything is logical and clear. One feature I feel missing is the geometry of the salt intrusion. Is the mixing zone front-like? Is the salt wedge is capped by river water? I do not think you need to complicate the text further by going into the physical explanation, but a 2D plot showing the distribution of salinity along the axis of the estuary at high tide and low tide would be informative.

* [464] haline stratification

* [467] Words are missing. Perhaps, you were writing “and to changes in river discharge”

* [468] Omit ‘important’

* [486] “the relationship between altitude and emergence time is not linear, because the latter factor also depends on the tidal range that follows lunar-solar fortnightly…”. The meaning is either vague or wrong. Please re-phrase and elaborate. Emergence time, I believe, depends on the form of the quasi-sinusoidal dynamics of the tide. If you mean that the tide range may vary along the estuary, than this meaning is obscured by inaccurate wording.

* [488] Please elaborate on “the tidal range that follows … yearly cycles”. I do not understand what you mean.

* [498-500] Consider omitting. All your readers know what spring and neap tides are.

* [494] I would much prefer you explain why you need a tidal coefficient and how you apply it. Or omit it.

* [501-503] I do not understand what you mean. I have re-read twenty times. Explain using other words, elaborate. What does it have to do with ‘micro- to macrotidal regime’?

* [512] “intrusion of the salt wedge becomes almost impossible.” You may name the river and may specify whether the phenomenon is year-round or occurs during the flood. 

* [525] What is ‘short longevity’? The term ‘turnover time’ is applicable to chemical compounds, but you speak of foraminifera and probably mean to say something like ‘generation time’ or ‘life cycle’

* [533-539] This consideration of fresh-water residence time is not linked in an obvious way to the topic of the section (Importance of fluvial discharge). Consider adding an explanation how fresh-water residence time controls the intrusion of the salt wedge into the estuary.

* [540] Modify the sentence to make it clear that you have invented and are introducing this calculation, that it is not taken from other publications.

* [544] Adding units to your formula will show that your Fluvial Influence value is a water-flow velocity (m s-1). Which means the formula actually compares the cross-section of the river at the gauge and the cross-section of the estuary at the sampling station. If the river flows at 3 m s-1, and the river/estuary cross-section ratio is 1:10, then Fluvial Influence (the velocity at which the thoroughly mixed estuarian water is calculated to progress seawards) is 0.3 m s-1. Consider supplying your formula with an explanation of this kind.

* [546-547]  This sentence contradicts to the idea of the section, which was introduced at the beginning of its first paragraph. This section is about blocking the intrusion of the salt wedge, not about salinity gradients. Consider omitting the sentence.

* [548-549] This is kind of obvious. The river/estuary cross-section ratio of course will vary between estuaries.  I think this paragraph needs to be appended with concluding remarks, where you explain, which Fluvial Influence velocities will prevent salt wedge intrusion.

* [587-618] The description of how equation 3 is constructed is rather complicated. I cannot say I am able to follow the explanation; normalization to the Loire river is particularly difficult for me. Anyway, I assume this description is all right. One critical comment is that the CS value (estuarine cross-section) is said to be measured at every station, but I do not see the method of measurement (if a description of the technique exists somewhere, it is not easy to find). Thus I do not understand how the RD/CS value shown for each station (tables 1,2,3,4) were obtained.    

* [595] the Loire river estuary is given quite some attention. A map would help. Alternatively, you could add an electronic supplementary material or provide the link to an internet project that describes the estuary.

* [629] If there are online image albums that show the setting, you could refer to them here. You could also upload photos taken during the fieldwork as electronic supplementary materials. 

* [633] What is the Auray river and how does it differ from the Auray river estuary? Where is the Armorican mountain chain and why does it need to be mentioned? What is the Morbihan Bay which is not shown on the map? Is this name relevant if it is never mentioned again? Does a river makes a part of a sea bay? Please speak of the estuary. When you say river instead, it confuses the reader. This and following text of section 6 needs work on clarity and should be shortened. 

* [675] What was monitored at these points and for how long?

* [682-709] The text is a bit chaotic. Please sort out the methods and results. The RD/CS values shown in the tables come from nowhere. Describe all techniques in the Methods paragraph or refer to the previous chapters where they are described. Describe the technique of measuring the estuarine cross-section at each station. Describe how the altitudes were measured. Were there altitude and time offsets between the reference point and the sampling stations?

* [fig 3] Take away the confusing blue arrow. Write MII=0.25 and MII=0.5 instead of 0.25 and 0.5 on the isolines. If possible, mark Mean Spring Tide and Mean Neap Tide. Label the subtidal channel. Same applies to fig 5.

* [706] This is obvious information. If you think you need to explain what the axes of the plot are, do it in the caption.

* [814] why is it ‘ria-like’? You said previously it was a ria. What do you mean by ‘an open marine bay’? I do not understand which part of your map shows an open marine bay.

* [822] Chapter 7 reads as Conclusions, not as Discussion and Conclusions.
